# Production and Immunogenicity of a Tag-Free Recombinant Chimera Based on PfMSP-1 and PfMSP-3 Using Alhydrogel and Dipeptide-Based Hydrogels

**DOI:** 10.3390/vaccines9070782

**Published:** 2021-07-13

**Authors:** Gaurav Anand, Saikat Biswas, Nitin Yadav, Paushali Mukherjee, Virander Singh Chauhan

**Affiliations:** 1Molecular Medicine Group, International Centre for Genetic Engineering and Biotechnology (ICGEB), New Delhi-110067, India; gauravanandicgeb@gmail.com (G.A.); saikat.biswas334@gmail.com (S.B.); nitin.acp@gmail.com (N.Y.); 2Multi Vaccines Development Program, International Centre for Genetic Engineering and Biotechnology (ICGEB Campus), New Delhi-110067, India; paushali.mukherjee@mvdp.org.in

**Keywords:** malaria, recombinant vaccine, fusion chimera, dipeptide, hydrogel, immunogenicity

## Abstract

A fusion chimeric vaccine comprising multiple protective domains of different blood-stage *Plasmodium falciparum* antigens is perhaps necessary for widening the protective immune responses and reducing the morbidity caused by the disease. Here we continue to build upon the prior work of developing a recombinant fusion chimera protein, His-tagged PfMSP-Fu24, by producing it as a tag-free recombinant protein. In this study, tag-free recombinant PfMSPFu24 (rFu24) was expressed in *Escherichia coli*, and the soluble protein was purified using a three-step purification involving ammonium sulphate precipitation followed by 2-step ion exchange chromatography procedures and shown that it was highly immunogenic with the human-compatible adjuvant Alhydrogel. We further investigated two dipeptides, phenylalanine-α, β-dehydrophenylalanine (FΔF) and Leucine-α, β-dehydrophenylalanine (LΔF) based hydrogels as effective delivery platforms for rFu24. These dipeptides self-assembled spontaneously to form a highly stable hydrogel under physiological conditions. rFu24 was efficiently entrapped in both the F∆F and L∆F hydrogels, and the three-dimensional (3D) mesh-like structures of the hydrogels remained intact after the entrapment of the antigen. The two hydrogels significantly stimulated rFu24-specific antibody titers, and the sera from the immunized mice showed an invasion inhibitory activity comparable to that of Alhydrogel. Easily synthesized dipeptide hydrogels can be used as an effective antigen delivery platform to induce immune responses.

## 1. Introduction

*Plasmodium falciparum* malaria is one of the main causes of the high morbidity and mortality among infants and children in tropical and sub-tropical countries [1]. Prevailing control measures such as anti-malarial drugs and insecticides have decreased the incidence of malaria infection and also reduced malaria transmission in many endemic areas. However, maintaining these gains is proving to be challenging and even reversible in some areas, with an increase in resistance to drugs and insecticides. The malaria vaccine is considered an important tool for fast-tracking the efforts to control the disease [2,3].

The RTS,S/AS01E, which targets the sporozoites, is the only malaria vaccine with regulatory approval. However, the efficacy data had shown a modest efficacy of 29–36% against falciparum malaria that waned over time and was less effective in infants compared to children [4]. Therefore, to achieve the target of eradication, better malaria vaccines will have to be designed as part of the integrated control strategy [2,5].

Since the erythrocytic stage of the parasite is responsible for the clinical disease, a vaccine targeting this stage would effectively reduce the morbidity and mortality due to *P. falciparum* [6]. Blood-stage vaccines primarily induce antibodies and prevent the invasion of erythrocytes by merozoites. Antibodies act by directly inhibiting a merozoite invasion of erythrocytes, antibody-dependent cellular cytotoxicity and complement lysis, and thus prevent the growth of parasites in RBCs. However, any progress in the development of blood-stage vaccines has proved to be difficult because most of the potential vaccine candidate antigens are polymorphic and also due to a considerable redundancy in erythrocyte invasion pathways [7,8].

Global efforts toward developing blood-stage malaria vaccines have identified several key targets that mediate erythrocyte invasion. More than 30 blood-stage malaria vaccine trials have been completed, with some targeting the antigens Merozoite Surface Protein 1 (MSP-1) and Apical Membrane Antigen-1 (AMA-1), and the remaining few targeting other antigens like Erythrocyte Binding Antigen-175 (EBA-175) and Merozoite Surface Protein 3 (MSP-3) [9,10]. MSP-1_19_ and AMA-1 have drawn maximum attention, as they both play an essential role in the invasion of erythrocytes, but failed to induce an optimal protection in field efficacy studies [11,12,13,14,15].

However, the failures of these single antigen-based blood-stage vaccines to elicit a protective effect do not completely discredit their vaccine potential. Considering that there is molecular redundancy associated with erythrocyte invasion, targeting multiple antigens will be necessary to limit the parasite’s ability to escape the host immunity and prevent clinical disease. A combination vaccine approach or a fusion protein chimera-based blood-stage vaccine may help to achieve adequate protective efficacy [16,17].

PfMSP-1, a major surface protein on merozoites, is involved in the erythrocyte invasion by the parasite [18,19,20,21]. It is speculated that PfMSP-1 mediates the initial attachment of merozoite with the RBC through the EGF-like domains of PfMSP-1_19_. Field studies and experimental challenge studies show that antibodies to PfMSP-1_19_ have significant invasion inhibitory activities and effectively inhibit parasite multiplication inside red blood cells [22,23,24,25]. Another merozoite surface antigen, PfMSP-3, exhibits parasite neutralization through a novel mechanism known as antibody-mediated cellular inhibition (ADCI) [26]. Immuno-epidemiological, preclinical and clinical studies revealed that anti-PfMSP3 antibodies correlated with protective immunity [27,28,29].

We have described earlier the development of a fusion protein, PfMSPFu24, comprising conserved regions of PfMSP-3 and PfMSP-1. The 19kDa region of PfMSP-1 (PfMSP-1_19_) was fused to the 11kDa region of PfMSP-3 (PfMSP-3_11_), that contains both T-helper (Th) epitopes and B cell epitopes that are targets of antibody-dependent cellular inhibition (ADCI) [30,31]. Our studies with the recombinant histidine-tagged MSPFu24 have shown that both protein fragments PfMSP-1_19_ and MSP-3_11_ retained their characteristic structure and antigenicity, and the immunization with the fusion protein elicited both invasion inhibitory antibodies and antibodies that inhibited parasite growth through ADCI activity]. Thus, the initial pre-clinical studies incorporating both target proteins in one construct appeared promising, with significant potential for further translational development.

Despite the advantages, recombinant subunit vaccines are weak immunogens, induce a poor immune response by themselves [32,33] and hence require an adjuvant and/or delivery system to improve immunogenicity [34,35,36,37]. The advantage of formulating a recombinant vaccine with an adjuvant is that it can reduce the amount of antigen in each dose and the number of vaccination doses.

Currently, aluminum salts (alum, Alhydrogel), MF59, ASOs and GLA-SE are a few vaccine adjuvants approved for human use [32,38]. There is much interest in the development of a safe and efficient vaccine delivery system, and nanotechnology has been widely applied to formulate a nanosized vaccine delivery platform that can promote immune responses [39,40]. Recently, the use of hydrogels—particularly those based on peptides—is being explored for the delivery of biomolecules, due to their intrinsic biocompatibility with various biological stimuli [41,42]. Self-assembled dipeptide hydrogels forming three-dimensional (3D) networks of fibrillar chains retain >90% of water molecules and have well-controlled biological, mechanical and material properties [43]. Additionally, they are easy to synthesize and characterize, and are also biocompatible and biodegradable. These advantages have allowed peptide hydrogels to become a successful biomolecular, specifically protein-based, antigen delivery platform [44].

We have been working with the development of short peptide-based nanostructures as a biomolecular delivery system [45,46,47,48,49] and have recently reported that the two ultra-short dipeptides, phenylalanine-α, β-dehydrophenylalanine (F∆F) and leucine-α, β-dehydrophenylalanine (L∆F), spontaneously self-assemble in physiological conditions into a 3D supramolecular network structure forming a highly fibrous structure at relatively low concentration [50,51]. Dipeptide hydrogels have evolved as a suitable delivery system for a wide variety of biomolecules including subunit vaccines [52].

The present study is divided into two parts. First, in view of the regulatory requirement, we have constructed and produced a tag-free recombinant PfMSPFu24 (rFu24) protein in an *Escherichia coli* expression system. In this study we report that the tag-free rFu24 protein retained the native conformation of the PfMSP-1_19_ component and induced invasion inhibitory antibodies in small animals as reported earlier.

Secondly, we developed and investigated the ability of two dipeptide-based hydrogels (F∆F and L∆F) as an antigen delivery system. In this study, we explored the potential of these short dipeptide-based hydrogels for the delivery of an immunogen-like tag-free rFu24 protein and compared it with traditionally used alhydrogels. Immunogenicity studies showed that the two peptide-based hydrogels may be suitable for the delivery of subunit-based immunogens.

## 2. Materials and Methods

### 2.1. Tag-Free rFu24 Cloning in pET-24b Vector

A synthetic gene encoding a codon-optimized PfMSPFu24 chimeric gene with C-terminal 6-His tag was described earlier [30,31]. The DNA fragment corresponding to PfMSPFu24 (1–172 amino acids), excluding the 6-His tag sequence (167–172), was amplified using the primers (Fwd-5′-TAT ACA TAT GGC AAA GAA TGC TTA CGA AAA GGC-3′; Rev–5′-CCC TCG AGT TAG GAG CTG CAG AAG ATA CC-3′) from the codon-optimized gene. The PCR product encoding rFu24 was digested with NdeI and XhoI (New England Biolabs, Beverly, MA, USA) and cloned into pET-24b (Novagen, San Diego, CA, USA) with a stop codon before the C-terminal 6-histidine (6-His) tag to obtain the plasmid rFu24-pET24b. The PCR amplified DNA fragments and the vector (pET-24b) were gel-extracted and digested with the NdeI/XhoI restriction enzymes. The digested inserts and vector were ligated overnight at 16 °C using a T4 DNA ligase (New England Biolabs, Beverly, MA, USA). The ligation mix was transformed into *E. coli* DH5-α competent cells, and positive clones containing the respective insert were confirmed by restriction digestion analysis.

### 2.2. Expression and Purification of the Full-Length rFu24

The resultant plasmid construct, rFu24-pET24b, was transformed into *E. coli* Rosetta-gami 2(DE3)pLysS competent cells (Novagen Inc., Madison, WI, USA). After screening the transformant colonies for the expression of rFu24, the selected colonies were grown in a semi-defined medium composed of 1.6% Select soyatone and 1.0% Yeast extract containing kanamycin (30 μg/mL) at 37 °C, shaking until the culture reached an optical density at 600 nm (OD_600_) of 0.6 to 0.7. The expression of recombinant Fu24 was induced by the addition of 0.8 mM isopropyl-β-D-thiogalactopyranoside (IPTG) (Sigma, St. Louis, MO, USA). The induced culture was further grown at 16 °C for 18 h and then harvested by centrifugation, and a cell pellet was stored at −70 °C. The *E. coli* cell pellet was thawed, re-suspended in a sonication buffer (20 mM Tris, pH 8.0, and 500 mM NaCl containing 1 mM benzamidine hydrochloride and 1% Tween 20) and sonicated at 4 °C using five sonication cycles, each comprising 10 s pulses at 10 s intervals. Bacterial lysates were centrifuged at 15,000× *g* for 30 min at 4 °C. The clear supernatant was precipitated with 30–40% *w*/*v* of ammonium sulphate at 4 °C for 2 h. The precipitated protein was harvested by centrifugation at 10,000× *g* for 30 min at 4 °C and re-suspended in an equilibration buffer (50 mM Tris buffer, pH 7.6). The recombinant proteins were further purified by anion exchange chromatography using Q-Sepharose resin (GE Healthcare, Chicago, IL, USA) and a AKTA FPLC system (GE Healthcare). Briefly, the Q-Sepharose resin was packed in a XK16/20 column (GE Healthcare), washed and equilibrated with 20 column volumes of the equilibration buffer (50 mM Tris buffer, pH 7.6). The protein sample was loaded onto the Q-Sepharose resin at a flowrate of 1 mL/min. After loading, the column was washed with 10 column volumes of the equilibration buffer. The bound protein was eluted from the column using a step gradient of NaCl (100 mM–500 mM) at a flow rate of 1 mL/min. The Q-Sepharose purified eluates were analyzed by SDS-PAGE. Fractions containing the recombinant protein were pooled, diluted 20-fold with a 50 mM sodium acetate–20 mM NaCl (pH 4.5) buffer and further purified by mixed mode cation exchange chromatography using a CMM HyperCel column (Cation Exchange mixed mode resin; PALL Lifesciences, Port Washington, NY, USA). Briefly, the CMM HyperCel resin packed in (XK16/20, column GE Healthcare) was equilibrated with 20 column volumes of the equilibration buffer (Sodium acetate–20 mM NaCl, pH 4.5). The recombinant protein sample was then loaded onto the column at a flowrate of 1 mL/min. After loading, the column was washed with 10 column volumes of the equilibration buffer, and the recombinant protein was eluted from the column using a pH gradient of sodium acetate (pH 4.5–pH 8) at a flow rate of 1 mL/min. All purification processes were carried out with buffers prepared using pyrogen-free water. Eluates were checked for purity by SDS-PAGE on a 15% acrylamide gel, and eluates with pure, monomeric rFu24 were pooled and concentrated using 3kDa molecular weight cut-off centricons (Millipore, Merck, Darmstadt, Germany). Next, the protein concentration was measured using a Pierce™ BCA Protein Assay Kit (Thermo Scientific, Rockford, IL, USA), filter-sterilized and stored at −80 °C until further use.

### 2.3. Characterization of Recombinant rFu24

rFu24 was characterized for purity by SDS-PAGE. The purified rFu24 was tested by Western blotting using rFu24-specific polyclonal antibodies raised in mice and used as reagent in Western blot to test antigenicity. Briefly, reduced and non-reduced rFu24 proteins were separated on 15% SDS-PAGE and transferred to a nitrocellulose membrane. After blocking with 5% skimmed milk in PBS at 37 °C for 2 h, the blot was first incubated with PfMSPFu24-specific polyclonal antibodies [30,31] and then with an HRP-conjugated anti-mouse IgG secondary antibody (Sigma, St. Louis, MO, USA) for 1 h. After washing, the bound protein in the immunoblot was detected with Pierce™ ECL Western Blotting Substrate (Thermo Scientific, Rockford, IL, USA) and 1 μL/mL H_2_O_2_ in PBS.

The conformation integrity of the PfMSP-1_19_ fragment in recombinant rFu24 was analyzed by an enzyme-linked immunosorbent assay (ELISA) using conformation specific monoclonal antibodies 1H4 under non-reducing and reducing conditions, as described earlier [30,31,53]. The homogeneity of rFu24 was determined by reverse phase high pressure liquid chromatography (RP-HPLC) using a C-18 column (Waters, Milford, MA, USA). The gradient profile was developed using buffer A (0.1% triflouroacetic acid in water) and buffer B (0.1% triflouroacetic acid in acetonitrile) as follows: 0 min, 95% Buffer A, 5% Buffer B; 45 min, 5% Buffer A, 95% Buffer B. Ellman’s test was used to detect the presence of any free thiol groups in the recombinant rFu24 protein. The rFu24 sample and different concentration of cysteine hydrochloride used as standard (ranging from 0 mM to 1.5 mM) were incubated with Ellman’s reagent (5′-dithio-bis-3-nitro benzoic acid) at room temperature for 15 min, and the absorbance was measured at 412 nm. Endotoxin levels in the protein sample were measured with a Litmus amebocyte lysates (LAL) gel clot assay (Charles River Endosafe). Host cell protein contamination in the rFu24 protein sample was analyzed by a commercially available ELISA kit (Cygnus Technologies, Southport, NC, USA).

### 2.4. Synthesis of Dipeptides

F∆F and L∆F were synthesized using a standard solution phase peptide synthesis method as described earlier [26,27] (Supplementary S1). The synthesized F∆F and L∆F were purified using reverse phase high performance liquid chromatography (RP-HPLC) (Shimadzu) on a C-18 column (Waters, Milford, MA, USA) with a 5–95% gradient of acetonitrile water containing 0.1% TFA (degassed and filtered). F∆F and L∆F were characterized using analytical reverse phase high-pressure liquid chromatography (RP-HPLC) using C-18 column and electron spray ionization mass spectrometry (ESI-MS).

### 2.5. Preparation of Hydrogel and Entrapment of Antigen

F∆F was dissolved in hexafluoroisopropanol (HFIP) at 50 mg/mL and L∆F in methanol at 50 mg/mL using 10 min sonication. Dipeptide hydrogels at 1 wt % were formed by adding 0.8 M sodium acetate buffer (pH 7) to the peptide solution at room temperature. The 1 wt % F∆F and L∆F gels were kept in a vacuumed desiccator for 3 h, and this was followed by heating at 50 °C–60 °C for 10 min to remove the residual amount of HFIP and methanol. Before adding the antigen to F∆F and L∆F, the gel suspensions were allowed to reach room temperature. Twenty-five micrograms of rFu24 were added to 1 wt % F∆F and L∆F hydrogels at a ratio of 1:50 (rFu24: F∆F/L∆F) by vortexing for 2 h at room temperature. After mixing, the dipeptide–antigen complexes were allowed to stabilize undisturbed for 1 h. The complexes were mixed again by vortexing for 10–15 min before being used for immunization.

### 2.6. Characterization of F∆F and L∆F Hydrogels

To determine the strength and stability of peptide hydrogels and antigen-loaded hydrogels, rheology experiments were performed on a Physica MCR 301 rheometer (Anton paar) using a parallel plate geometry tool with a diameter of 25 mm. F∆F and L∆F (at 1 wt % concentration) and the loading of rFu24 to 1 wt % F∆F and L∆F were prepared and immediately transferred to the rheometer’s bottom plate. An amplitude sweep (0.01–100% strain) experiment was performed at a constant oscillatory frequency of 1Hz, and storage modulus (G′) and loss modulus (G″) values were recorded.

The size and morphology of the F∆F, L∆F, F∆F + rFu24 and L∆F + rFu24 hydrogels were studied using transmission electron microscopy (TEM). Twenty microliters of F∆F and L∆F gels (1 wt %) were loaded onto carbon-coated copper grids for 45 min at room temperature, and then the hydrogels were negatively stained with 1% uranyl acetate. The grids were visualized using the 120 kV mode of TEM (Tecnai 12 BioTWIN, FEI, Eindhoven, Netherlands), and the micrographs were analyzed using the Analysis II (Megaview, SIS, Münster, Germany) software.

### 2.7. Cell Viability Experiments

The effect of F∆F and L∆F, on the viability of Huh 7 and HepG2 cells was tested using a standard 3-(4,5-dimethylthiazolyl-2)-2,5-diphenyltetrazolium bromide (MTT) assay. Both cell lines were grown to confluency in DMEM culture media supplemented with 10% fetal bovine serum and 0.1% penicillin-streptamycin at 37 °C, 5% CO_2_. As described above, 1 wt % dipeptide gels of F∆F and L∆F hydrogels were prepared. To allow the evaporation of HFIP and methanol, the hydrogels were kept overnight inside the culture hood in a vacuumed desiccator. The hydrogels were then washed three times each with PBS and an incomplete medium. Before seeding, the hydrogels were given one final wash with complete media. Subconfluent Huh 7 and HepG2 cells were harvested by trypsinization and washed with PBS and complete media. Cells at a concentration of 2 × 10^3^ cells/well were plated in a 96-well plate and incubated with 1 wt % F∆F and L∆F hydrogels for 72 h. In the control group, cells were treated with the culture medium only. The cell viability of treated and untreated cells was measured by an MTT assay. Absorbance was measured on a microplate reader VersaMax ELISA reader (Molecular Devices, Sunnyvale, CA, USA) at 570 nm and cell viability (%) was calculated using the equation: (Absorbance of treated cells/Absorbance of control) × 100.

### 2.8. Immunization of Mice with rFu24

BALB/c mice obtained from the Jackson Laboratory were housed in small-animal facilities of the International Centre for Genetic Engineering and Biotechnology (ICGEB) under pathogen-free conditions. All procedures were approved by the Institutional Animal Care and Use Committee (IAEC) (IAEC Approval Number: ICGEB/IAEC/02042019/MM-4); dated 02/04/2019. Groups of 6 to 10-weeks-old female mice (n = 6) were immunized by intramuscular route with 25μg of rFu24 formulated with Freund’s adjuvant, Alhydrogel^®^, FΔFloaded with rFu24 and LΔF-loaded with rFu24. The mice were immunized on days 0, 28 and 56. The vaccine formulations were made immediately prior to use. For immunization with the adjuvant formulation, a 26-gauge needle was used, and for hydrogel injections a 24-gauge needle was used. Freund’s adjuvant was used as a positive control in this study. Serum samples from each mouse was collected before immunization, at −1 day and three weeks after the third immunization, at day 70.

### 2.9. Ethics Statement

Animal studies were conducted in accordance with the guidelines of the Institutional Animal Care and Use Committee (IAEC), Department of Biotechnology, Government of India. ICGEB is licensed to conduct animal studies for research purposes under the registration number 18/1999/CPCSEA (dated 10 January 1999). All experimental protocols were reviewed and approved by the ICGEB Institutional Animal Ethics Committee (IAEC) (IAEC Approval Number: ICGEB/IAEC/02042019/MM-4); dated 2 April 2019.

### 2.10. Enzyme-Linked Immunosorbent Assay (ELISA)

Sera collected from immunized mice were tested for antibody titer specific for rFu24, PfMSP-1_19_ and PfMSP-3_11_ by ELISA. Flat-bottom 96-well, microtiter ELISA plates (MaxiSorp, Nunc) were coated with rFu24 (2 µg/mL), PfMSP-1_19_ (5 µg/mL) or PfMSP-3_11_ (2 µg/mL) in a carbonate-bicarbonate buffer overnight at 4 °C. After washing three times with phosphate-buffered saline (PBS) containing 0.05% Tween 20 (PBST), the plates were blocked with 2% skimmed milk powder in PBS (pH 7.2) for 2 h at 37 °C. One hundred microliters of serially three-fold-diluted immune sera in PBST containing 0.25% skimmed milk powder were added to antigen-coated plates and incubated for 1 h at 37 °C. The plates were washed 3 times with PBST, and bound antibodies were detected with goat anti-mouse IgG conjugated with horseradish peroxidase (Sigma) for 1 h at 37 °C. Finally, the ELISA plates were washed, and 1mg/mL o-phenylene diaminedihydrochloride (Sigma) as chromogen and hydrogen peroxide as substrate were added to each plate. The reaction was stopped with 2.0 N sulfuric acid. The absorbance was measured at 490 nm using a VersaMax ELISA reader (Molecular Devices, Sunnyvale, CA, USA). End-point titers were determined as the mean plus three standard deviations of absorbance of pre-immune sera. The end-point titer was calculated using a 4 parameter curve fitting model (GraphPad Prism Software, version 6.01, San Diego, CA, USA).

### 2.11. Invasion Inhibition Assay

Sera collected from immunized mice were tested for their ability to inhibit *P*. *falciparum* 3D7 replication by an in vitro growth inhibition assay (GIA), as described previously [30,31,54]. *P. falciparum* parasites were cultured in human O+ RBCs as previously described by Trager and Jenson [55]. For the parasite culture, human RBCs were procured from Rotary Blood Bank, New Delhi, India. For the invasion inhibition assay, synchronized parasites were adjusted to 2% hematocrit and 0.3% parasitemia in a late-trophozoites/early-schizonts stage and incubated in the presence or absence of sera. Pre-immune and immune sera were thawed at 37 °C in a water bath and heat-inactivated for 20 min at 56 °C. The sera were added to the parasite culture at a final dilution of 1:10 and incubated for 44 h under a mixed gas environment. Parasite growth was assessed by staining parasite-infected RBCs with ethidium bromide (Invitrogen Corporation, Frankfurt, Darmstadt, Germany) at a concentration of 10 μg/mL. After staining, RBCs were washed with PBS to remove excess dye, and 100 µL of lysis buffer were added and incubated for 10 min. Ethidium bromide fluorescence was measured at 485 nm excitation and 535 nm emission in the Versamax ELISA reader (Molecular Devices). The inhibition rate was determined as: % inhibition = [(1 − (Parasitemia of test IgG − Parasitemia of starting culture)/(Parasitemia of pre-immune IgG-Parasitemia of starting culture)] × 100%.

### 2.12. Statistics

The statistical analysis was done using the GraphPad Prism Software, version 6.01, San Diego, CA, USA. The results were described as mean ± standard error. Differences between the groups were compared using a one-way analysis of variance (ANOVA) followed by a Tukey’s post hoc analysis for a multiple comparison test. A *p* value < 0.05 was considered as statistically significant.

## 3. Results

### 3.1. Expression, Purification and Characterization of 6-His Tag Free PfMSPFu24 (rFu24)

The PfMSPFu24 chimeric gene [30] construct excluding the 6-His tag sequence was cloned into a pET-24b vector, and the recombinant tag-free Fu24 (rFu24) protein was expressed in *E. coli* Rosetta-gami 2(DE3) cells as a soluble protein. rFu24 was purified from the cytosolic fraction by ammonium sulphate precipitation, followed by anion-exchange chromatography and mixed mode cation-exchange chromatography (Figure 1a,b).

Purified rFu24 was observed as a single band on SDS-PAGE and by western blot under reducing and non-reducing conditions (Figure 1c,d, Appendix A). Quantitative densitometry analysis of immunoblot using Image Lab^TM^ showed that the intensity of non-reduced and reduced bands of rFu24 were 17,080.24 ± 909.94 and 13,970.39 ± 540.32, respectively (Appendix A). rFu24 contains 12 cysteine residues located in MSP-1_19_ to form the disulfide bonds (30,31). The observed difference seen in the mobility of rFu24 on SDS-PAGE gels under reducing and non-reducing conditions indicates the presence of intramolecular disulfide bonds. The purity of rFu24 was analyzed by reversed phase high-performance liquid chromatography (RP-HPLC). rFu24 eluted as a single symmetric peak with more than 98% purity (Figure 2a).

In the fusion chimera it is important to maintain the conformational integrity of immunologically relevant PfMSP-1_19_. The conformational integrity of the PfMSP-1_19_ component in rFu24 was tested with 1H4, a PfMSP-1_19_-specific conformation-sensitive monoclonal antibody, by ELISA. Native rFu24 showed a strong reactivity with 1H4, confirming that the conformational epitopes of the native PfMSP-1_19_ were intact in rFu24 (Figure 2b), and this reactivity of rFu24 with 1H4 declined to an insignificant value under denatured conditions, further confirming that the integrity of PfMSP-1_19_ in the fusion chimera remained undisturbed. The presence of any free thiol group in the purified rFu24 was analyzed by an Ellman test. The negative test results indicated that there were no free cysteines present in rFu24. The purified rFu24 was evaluated by an ELISA using polyclonal antisera against the full-length PfMSP-3 and PfMSP-1_19_. Both the native and denatured forms of rFu_24_ reacted strongly with anti-PfMSP-1_19_ and anti- PfMSP-3 antibodies (Figure 2b). The endotoxin content in rFu24 was measured by the Limulus amebocyte lysate (LAL) test. The endotoxin level in the purified rFu24 was less than 5 EU per 25 µg of rFu24.

### 3.2. Synthesis and Characterization Dipeptide F∆F and L∆F Hydrogel

A standard solution phase synthesis procedure was used to synthesize F∆F and L∆F as described earlier [46,47,48]. Peptides were characterized by RP-HPLC and mass spectrometry (Appendix A). The hydrogels FΔF and LΔF were spontaneously formed at 1 wt % peptide concentration and were observed to be colorless and translucent. The tube inversion test was carried out, and it was found that both gels did not flow down and thus were found to be self-supportive. (Figure 3a, Appendix A). The ultrastructures of the hydrogels were investigated using TEM. The electron micrographs showed that both the F∆F and L∆F hydrogels had a nanofibrillar 3D mesh-like structure (Figure 3b,d). Next, we measured the viability of the Huh 7 and HepG2 cell lines after culturing the cells on 1 wt % gel by an MTT assay, to assess the biocompatibility of F∆F and L∆F. The Huh 7 cells showed 95.3 ± 2.36% and 98.9 ± 0.87% viability with F∆F and L∆F hydrogels respectively, whereas HepG2 cells showed 98 ± 0.77% and 94.0 ± 0.07% % viability with F∆F and L∆F hydrogel, respectively (Figure 3f,g). Thus, both the F-∆F and L∆F hydrogels showed no cytotoxicity at 1 wt % gels.

### 3.3. Entrapment of rFu24 and Its Rheological Characterization

To study antigen entrapment in the hydrogels, 25 μg of rFu24 were added to 1 wt % F∆F and L∆F hydrogels at a ratio of 1:50 (rFu24: F∆F/L∆F) and incubated at room temperature for one hour. To check the entrapment of rFu24 in the two gel matrices, supernatants were collected and checked by SDS-PAGE. rFu24 was efficiently entrapped in both the F∆F and L∆F hydrogels, and no free rFu24 was observed in the supernatant of F∆F-rFu24 and L∆F-rFu24 gels (Figure 4a). TEM images showed that the 3D mesh-like structures of the hydrogels remained intact after the entrapment of rFu24 (Figure 3c,e).

We also carried out rheological experiments in order to measure the mechanical strength of two hydrogels at 1 wt %. The storage modulus (G′) values, measuring the mechanical strength, were 520 ± 77.58 kPa for F∆F and 30.86 ± 0.47 kPa for L∆F. The mechanical strength of the complexes of F∆F and L∆F after entrapping rFu24 at 1 wt % decreased to 263 ± 65.83 kPa and 19.63 ± 3.0 kPa, respectively (Figure 4b,c).

### 3.4. Immunogenicity of rFu24

To evaluate the immunogenicity of rFu24, six BALB/c mice per group were immunized intramuscularly three times at a four-week interval with rFu24 formulated with different antigen delivery systems as per the schedule shown in Figure 5a. rFu24 was formulated with aluminum hydroxide (Alhydrogel^®^), the two dipeptide hydrogels (F∆F-rFu24 and L∆F-rFu24 gels) and the Freund’s adjuvant which was used as control adjuvant. Serum samples were collected prior to immunization and three weeks after the last immunization and tested by an ELISA for antibody responses specific to rFu24 and also to the PfMSP-1_19_ and PfMSP-3_11_ components of the chimera.

Sera from mice immunized with rFu24 with different adjuvant formulations (Freund’s, Alhydrogel, F∆F and L∆F) showed high antibody titers against all the three antigens—rFu24, PfMSP-1_19_ and PfMSP-3_11_—while no antigen-specific antibody responses were found in pre-immune sera.

The Freund’s adjuvant control group induced high antibody titers against rFu24 (geometric mean, 1 × 10^6^), PfMSP-1_19_ (geometric mean, 7.8 × 10^5^) and PfMSP-3_11_ (geometric mean, 4.1 × 10^5^). This was followed by the Alhydrogel group, which was less immunogenic and induced a lower antibody response against rFu24: (geometric mean, 1.5 × 10^5^) PfMSP-1_19_ (geometric mean, 1.2 × 10^5^) and PfMSP-3_11_ (geometric mean, 1.1 × 10^5^). rFu24 formulated with F∆F hydrogel induced a higher antibody titer against rFu24 (geometric mean, 1.1 × 10^5^), PfMSP-1_19_ (geometric mean, 0.7 × 10^5^) and PfMSP-3_11_ (geometric mean, 0.6 × 10^5^) compared to L∆F hydrogel groups (rFu24: geometric mean, 6.6 × 10^4^; PfMSP-1_19_: geometric mean, 5.5 × 10^4^ and PfMSP-3_11_, geometric mean, 5.0 × 10^4^).

The rFu24 formulated with Freund’s adjuvant was found to be significantly immunogenic and induced high antibody titers against all three antigens compared to the Alhydrogel, F∆F and L∆F groups (*p* < 0.01, one-way ANOVA with Tukey’s multiple comparison post-test) (Figure 5b).

Antibody responses to rFu24, PfMSP-1_19_ and MSP-3_11_ in the Alhydrogel and F∆F hydrogel groups was more than the L∆F hydrogel group, however, there was no significant differences in the antibody titer between Alhydrogel and dipeptide-based hydrogels groups (*p* > 0.05, one-way ANOVA with Tukey’s multiple comparison post-test).

We also compared the immunogenicity of tag-free rFu24 with (6-His) tagged rPfMSP-Fu_24_ protein in Freund’s adjuvant, and no statistically significant difference was seen in the antibody titer between the tag-free and his-tagged chimera fusion proteins (Appendix A), indicating that the removal of His-tag has no effect on the immunogenicity of the chimera protein.

Next, we tested the invasion inhibitory activity of the antibodies present in the immune sera collected at day 70 from all groups by performing a growth inhibition activity (GIA) assay on *P. falciparum* 3D7. Immune serum samples from different adjuvant groups, when tested at a 1:10 dilution, were able to block the erythrocyte invasion by the parasite with varying degrees of efficacy (Figure 5c). Freund’s adjuvant group showed high inhibitory activity by reducing parasite growth by 55.4 ± 13.2% (Figure 5c). Immune sera from F-∆F-rFu24 showed inhibitory activity by reducing parasite growth by 42.4 ± 10.2%. In contrast, immune sera from alhydrogel and L-∆F-rFu24 showed less inhibitory activity (36.4 ± 10.2% and 32.4 ± 10.2%, respectively). However, there was no statistically significant difference in the efficiency of inhibition by sera from these different adjuvant groups. In summary, these immunogenicity results suggest that F∆F hydrogels can deliver antigens and elicit antibodies which partially inhibit parasite growth in vitro.

## 4. Discussion

We had described earlier the design and production of a fusion chimera, PfMSPFu24, comprising key parts of two major vaccine target antigens, PfMSP-1 and PfMSP-3. The fusion protein which contained the 6x-Histidine tag to assist in purification retained the immunologically important native conformational epitope of PfMSP-1_19_, showed enhanced immunogenicity compared to its individual components, PfMSP-1_19_ and PfMSP3_11_, and generated strain-transcending anti-parasitic antibodies [30,31]. To fulfill the current regulatory criteria for the development of subunit vaccines, we need to produce the fusion chimera without any tag and confirm its immunogenicity. In the present study we described the production of tag-free rFu24 and its immunogenicity using a human-compatible adjuvant and also two short peptide-based hydrogels as the antigen delivery system.

Earlier we had expressed recombinant 6x-His-tagged PfMSP-Fu24 in *E. coli* SHuffle cells as a soluble protein under cGMP specifications for vaccine development [31]. Recombinant Fu24 was expressed in *E. coli* Rosetta Origami DE3 cells as a soluble protein and was purified by precipitation using ammonium sulfate followed by two-step ion exchange chromatography. rFu24 was purified to homogeneity, as confirmed by a single main band on SDS-PAGE gel and a single symmetric peak in reverse-phase chromatography (RP-HPLC).

One main requirement of the chimera fusion protein is that the recombinant forms of each component should be in proper conformation and should induce functional antibodies against the native parasite proteins. We have shown that the rFu24 retained proper conformation of PfMSP-1_19,_ which is one of the component of the fusion chimera. rFu24 showed a significant shift in mobility between non-reduced and reduced proteins, indicating the presence of disulfide bonds. The 19 kDa fragment of PfMSP-1 folds into 2 EGF-like domains stabilized by six disulfide bonds [56]. A proper disulfide bond formation is essential not only for the native structure of PfMSP-1_19_ but also for inducing appropriate immune responses [24,56,57].

Moreover, rFu24 retained conformational epitopes of native PfMSP-1_19_, which was further confirmed through the recognition of the protein by conformational specific monoclonal antibodies. Our results show that the conformational monoclonal antibody 1H4 did not react with the denatured recombinant Fu24. This suggests that rFu24 retained the conformational integrity of PfMSP-1_19_ fragment, since this is essential for producing invasion inhibitory antibodies. The presence of an MSP-3 fragment, MSP-3_11_, in rFu24 was also confirmed by an ELISA using PfMSP-3_11_-specific polyclonal antibodies available from our earlier studies [30,31].

Subunit-based vaccines are generally less immunogenic and require the use of adjuvants or novel delivery systems to increase their immunogenicity and direct the type of immune responses for optimal efficacy [32,34]. Due to safety and tolerability concerns, there are only a limited number of licensed adjuvants approved for human vaccines [39,40]. There remains an urgent need to develop a new adjuvant formulation and/or an appropriate delivery system with sufficient potency and low enough toxicity for clinical use for the development of subunit vaccines.

Recently, hydrogels, particularly peptide-based, are being considered as a novel delivery system for biologics, including immunogens [58,59,60]. Peptide hydrogels offer several advantages, such as their easy synthesis and characterization, suitable mechanical properties, easy antigen entrapment, high biocompatibility and modulate immune responses. We investigated whether two conformationally restricted dipeptide hydrogels (FΔF and LΔF) can be used as effective delivery for recombinant Fu24 [49]. We have previously shown that under physiological conditions these dipeptides self-assemble spontaneously, form a highly stable hydrogel and allow for the incorporation of drugs by simple mixing [49,51].

FΔF and LΔF hydrogels were considered separately for the delivery of rFu24 for immunization. Our results confirmed the successful entrapment of rFu24 by the F∆F and L∆F hydrogels through simple mixing. TEM images confirmed that after the entrapment of rFu24 into both hydrogels, there were no significant morphological changes in the 3D fibrillar nanostructure of the gels [50,51,52]. We had earlier reported that 1 wt % hydrogels showed adequate mechanical strength, and may be used as delivery vehicles for rFu24 [51]. The results of rheological studies showed that the mechanical strength of the two hydrogels decreased after entrapping rFu24. Similar results have been reported for other hydrogels [61,62,63]. The entrapment of antigens can either increase or decrease the elasticity of the hydrogel, which is indicated by the higher or lower storage modulus (G′) values, and which in turn modulates the capacity of the hydrogels for sustained antigen delivery [61,62,63]. Controlling the mechanical strength of hydrogels can improve the cellular uptake and biocompatibility [64].

An indispensable property of any delivery system including peptide-based hydrogels is its biocompatibility. Our results showed that 1 wt % gel of F∆F and L∆F did not affect the viability for the Huh 7 and HepG2 cell lines, further suggesting the suitability of dipeptide hydrogels for in vivo delivery platforms for immunogens.

We assessed the antibody responses induced by rFu24 formulated with different adjuvants as an indicator of potential efficacy. It is worth noting that there was no difference between the immunogenicity of tag-free rFu24 and that of histidine-tagged PfMSPFu24. Clearly, the removal of His-tag has no consequence on the immunogenicity of the chimera protein.

The formulation of rFu24 with F∆F and L∆F hydrogels elicited a robust antigen-specific antibody response for the antigens rFu24, PfMSP-1_19_ and PfMSP-3_11_, when tested individually by ELISA. While the antibody response induced in the F∆F formulation was comparable to that of the human-compatible Alhydrogel adjuvant, the antibody titers of the L∆F formulation were lower than those of the F∆F and Alhydrogel groups. The most remarkable finding of the study was that the two hydrogels, F∆F and L∆F, seem highly suitable for the delivery of vaccines in vivo.

Recombinant antigens like PfMSP-1_19_ are known to be poorly immunogenic and unable to induce protective antibodies, as they lack effective CD4^+^ T cells capable of providing adequate help to B cells [65]. As reported earlier [30,31], in rFu24 this was overcome by the presence of T helper cell epitopes of PfMSP-3_11_, thereby enhancing the immunogenicity of PfMSP-1_19_. Thus, T cells restricted to the PfMSP-3_11_ domain distinctly provided adequate help for the activation and differentiation of both PfMSP-1_19_- and MSP-3_11_-specific B for the production of functional antibodies [30,31].

As stated previously, inducing invasion-inhibitory antibodies against parasite antigens is critical for the development of an effective blood-stage malaria vaccine. We demonstrated that the antibodies induced by rFu24 showed invasion inhibitory activities in the in vitro GIA. The invasion inhibitory activity of anti-rFu24 antibodies from the F∆F group was able to effectively reduce parasite growth as compared to the L∆F and Alhydrogel groups. These data clearly exemplified the presence and specificity of the growth-inhibitory antibodies induced by rFu24 formulated with peptide-based hydrogels. These hydrogels compared well with other peptide-based hydrogels reported as antigen delivery system and immunostimulatory agents [60,66,67]. The hydrogels described in this study are more attractive, since they are based on a dipeptide structure, also easy to make and characterize.

These results reconfirm that his-tag free rFu24 can be developed as part of an asexual blood-stage multivalent subunit vaccine. We have been able to develop a procedure to make tag-free recombinant Fu24 in the *E. coli* expression system. Generally, it has not been easy to produce recombinant proteins without any synthetic tag with satisfactory yields. In this context, it is quite remarkable that we were able to produce synthetic tag-free rFu24 with reasonable yields and high purity.

Of great interest is our finding that the two ultra-short peptide-based hydrogels, F∆F or L∆F, appear to be suitable as an effective delivery platform for subunit vaccines and as immunostimulatory agents. These hydrogels not only entrapped fusion chimera protein effectively but also induced antigen-specific humoral immune responses comparable to those of Alhydrogel. Dipeptides could be synthesized easily and in large scale by standard solution-phase peptide synthesis. Our findings establish the possibility of ultra-short peptide-based hydrogels being further developed as a delivery system for subunit vaccines like recombinant malaria vaccine candidates.

## 5. Conclusions

We have successfully developed a process for expression, purification, and characterization of rFu24 protein without the histidine tag. Formulations of rFu24 with Alhydrogel as well as two novel dipeptide-based hydrogels produced significant functional invasion inhibitory antibody responses. rFu24 may further be developed as a stand-alone or a component of a malaria vaccine. Likewise, the dipeptide-based hydrogel systems of F-∆F and L-∆F may further be developed as delivery systems for subunit vaccines.

## Figures and Tables

**Figure 1 vaccines-09-00782-f001:**
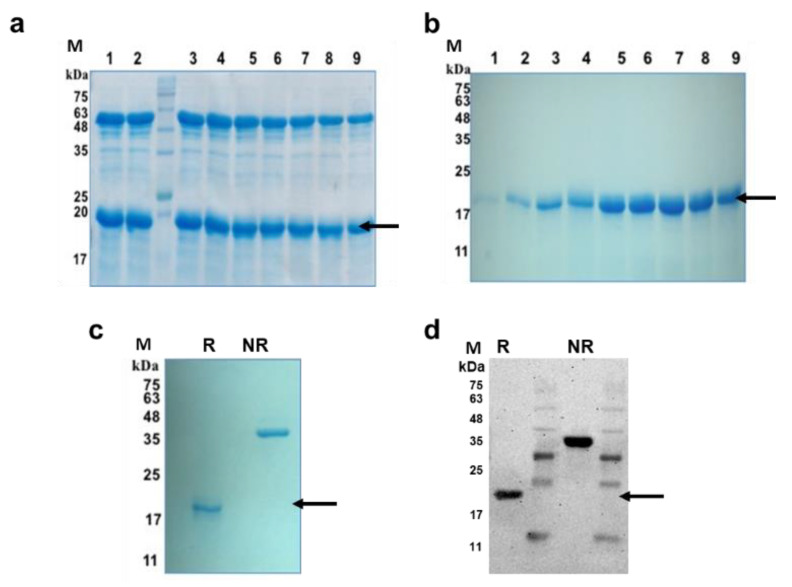
Purification of rFu24 protein. (**a**) Purification of rFu24 by anion exchange chromatography using Q-Sepharose. Lanes 1–9: Ion exchange purified eluates of rFu24; (**b**) Purification of rFu24 by cation exchange chromatography using CMMC. Lanes 1–9: Mixed mode cation exchange purified eluates of rFu24; (**c**) rFu24 purity was analyzed by 15% SDS-PAGE gel under reducing and non-reducing conditions and stained with Coomassie blue. (**d**) Immunoblot of purified rFu24 protein detected with anti-PfMSPFu24 polyclonal mouse serum under reducing and non-reducing conditions. The arrow indicates rFu24. M, molecular weight marker; NR, non-reduced; R, reduced.

**Figure 2 vaccines-09-00782-f002:**
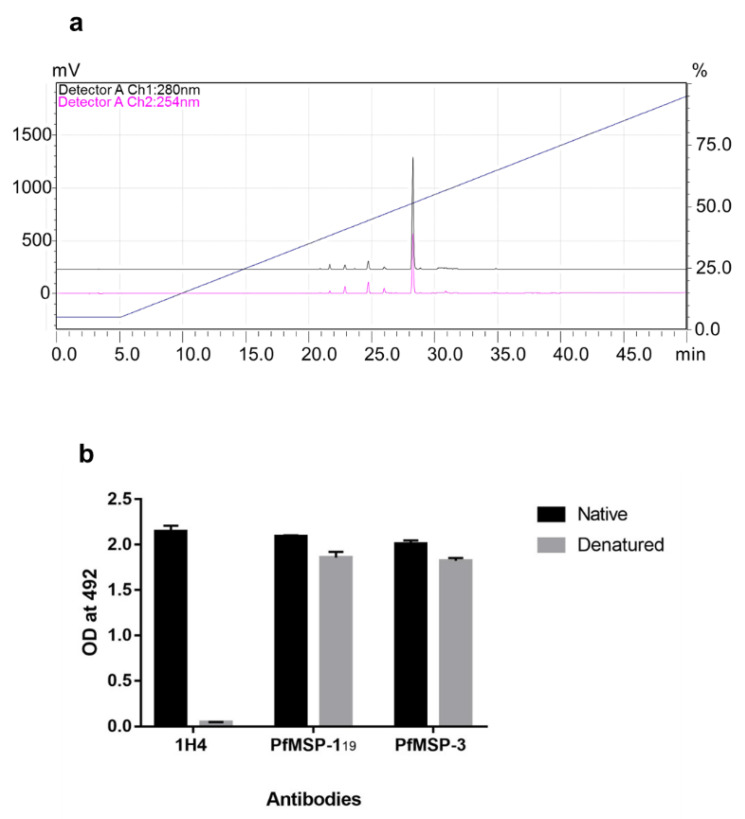
Characterization of purified rFu24 protein. (**a**) The purity of rFu24 was analyzed by RP-HPLC. Purified rFu24 eluted as a single symmetrical peak representing monomeric rFu24; (**b**) The conformation integrity of rFu24 under reducing and non-reducing conditions was tested with PfMSP-1_19_ conformation-specific Mab 1H4 by ELISA. rFu24 was also probed with polyclonal anti-PfMSP-1_19_ and PfMSP-3 mouse sera at a 1:2000 dilutions.

**Figure 3 vaccines-09-00782-f003:**
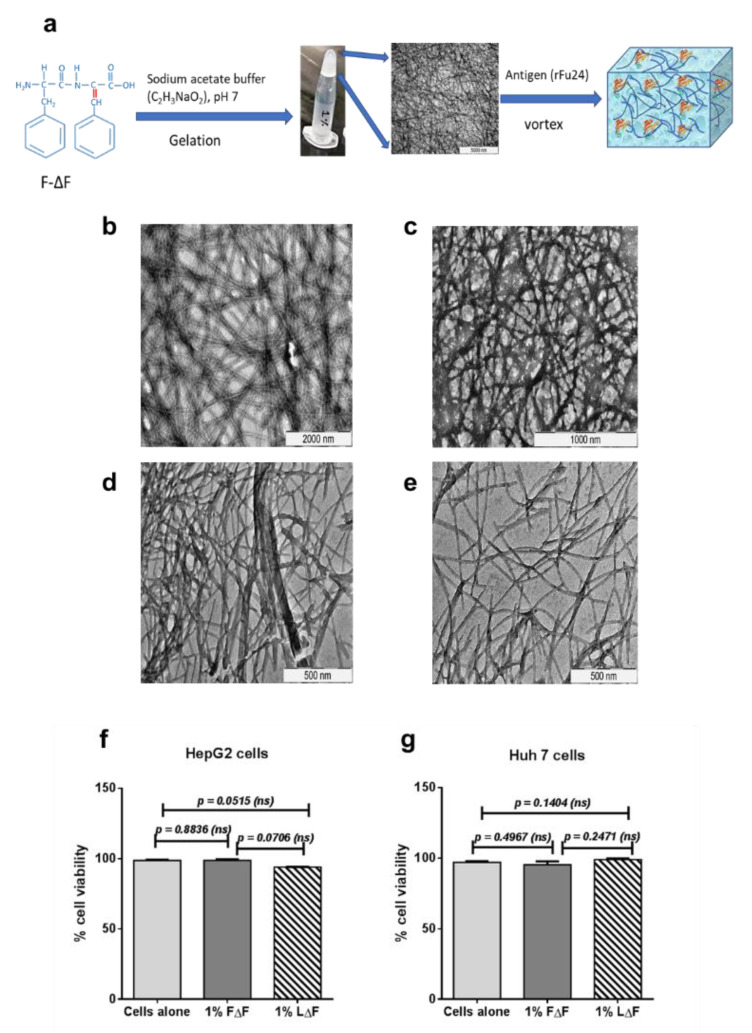
Characterization of F∆F and L∆F hydrogels. (**a**) Schematic diagram of hydrogel formation followed by the loading of rFu24. (**b**,**c**) Transmission electron micrographs of LΔF and FΔF hydrogels; (**d**,**e**) TEM after adding rFu24 to LΔF and FΔF hydrogels; (**f**) Cell viability of HepG2 cell line after incubation with FΔF and LΔF hydrogels for 72h; (**g**) Cell viability of Huh 7 cells after incubation with FΔF and LΔF hydrogels for 72 h.

**Figure 4 vaccines-09-00782-f004:**
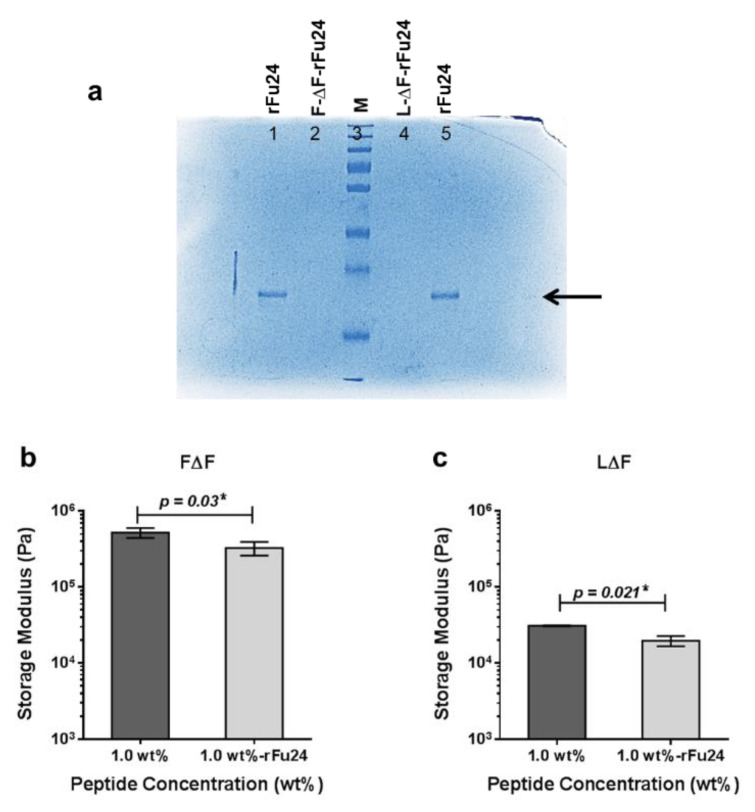
Entrapment of Fu24 and its rheological characterization (**a**) rFu24 was entrapped onto 1 wt % F∆F and L∆F hydrogels at the ratio of 1:50 (rFu24: F-∆F/L∆F). The entrapment of rFu24 by FΔF (FΔF-rFu24) and LΔF (LΔF-rFu24) is shown by SDS-PAGE. Lane 1 represents 5 µg of rFu24; lane 2 represents FΔF-rFu24; lane 3 represents the molecular weight marker in kilodaltons (kDa) (M); lane 4 represents LΔF-rFu24; and lane 5 represents 5 µg of rFu24. Mechanical strength of FΔF, LΔF, FΔF-rFu24 and LΔF-rFu24 hydrogels. (**b**) Storage modulus (G′ values) of FΔF and FΔF-rFu24 at 1 wt % and (**c**) LΔF and LΔF-rFu24 at 1 wt %. Graphs represent mean ± standard deviation (*n* = 3). * significant at *p* < 0.05.

**Figure 5 vaccines-09-00782-f005:**
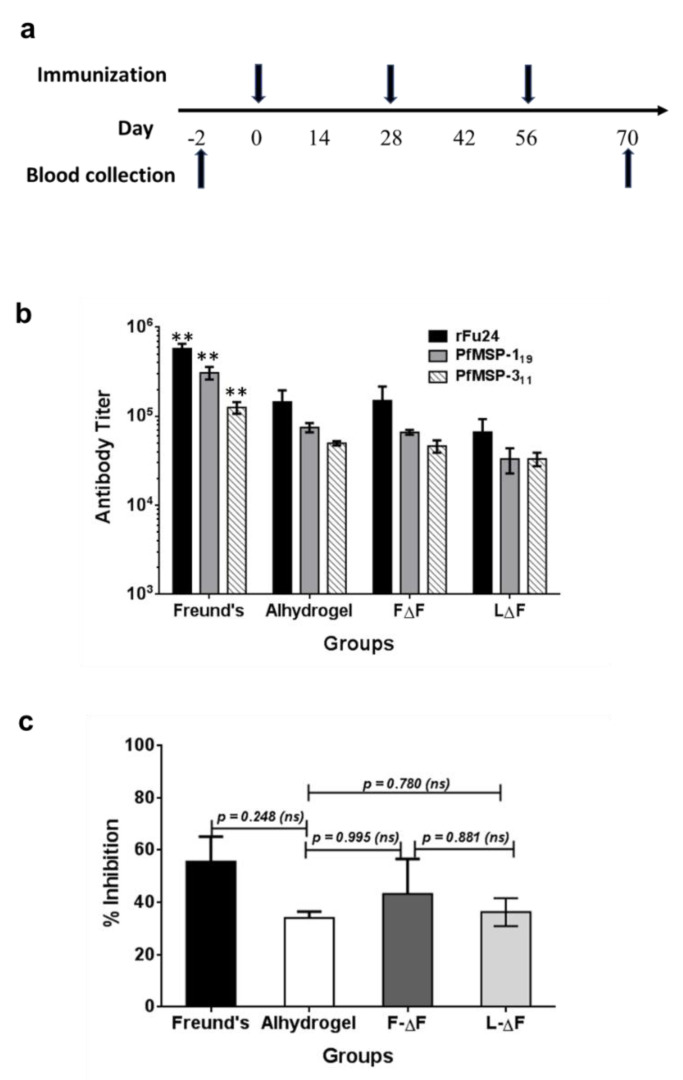
Immunogenicity and efficacy of rFu24. (**a**) Immunization scheme. (**b**) Groups of BALB/c mice (*n* = 6) were immunized three times with 25 μg/dose of rFu24 formulated with Freund’s adjuvant, Alhydrogel and dipeptide hydrogels, FΔF and LΔF. Sera collected following the final immunization were analyzed for antigen-specific IgG titers. The end-point titers (means ± SEM) against rFu24 were measured by ELISA. ** *p* < 0.01 (one-way ANOVA followed by Tukey’s multiple comparison test). (**c**) The inhibition of parasite invasion in vitro by anti-rFu24 antibodies was assessed against the P. falciparum 3D7 strain. Sera collected before and after the final immunization with rFu24 formulated with Freund’s adjuvant, Alhydrogel and the FΔF and LΔF hydrogels were tested at 1:10 dilution for the inhibition of erythrocyte invasion by parasites. Percent growth inhibition (shown as means and standard deviations) was calculated by using the parasitemia of the culture grown in the presence of test IgG with the parasitemia of the culture grown in the presence of preimmune IgG. *p* > 0.05; not significant (ANOVA followed by Tukey’s multiple comparison test).

## Data Availability

The data presented in this study are available in this article or its Appendix A.

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
