# Peer review of "Production and Immunogenicity of a Tag-Free Recombinant Chimera Based on PfMSP-1 and PfMSP-3 Using Alhydrogel and Dipeptide-Based Hydrogels"

_vaccines, 2021, doi:10.3390/vaccines9070782_

Round 1

Reviewer 1 Report

The paper of Gaurav Anand et al deals with the production of plasmodium falciparum vaccine. The authors report the production of a recombinant protein combinant MSP-1 and MSP-3 proteins. Using E coli, they produce and purify rFU24 that keep conformational integrity which is essential for producing invasion inhibitory antibodies. 

The authors use peptide-based hydrogels as antigen delivery system, two dipeptide-based hydrogels were evaluated , FΔF and LΔF. Both dipeptides successfully entrap rFu24 and do not induce cellular death in vitro.

In vivo, the intra-muscular injection of rFu24 formulated in FΔF and LΔF hydrogels induce high levels of antigen-specific rFU24 antibodies and can block parasite invasion of red blood cells in vitro.

Questions

  1. In order to better evaluate the antibody response elicitated by rFu24 formulated in FΔF and LΔF, can the authors also used MSP1 and MSP3 in their ELISA test ?
  2. In order to better evaluate the antibody response elicitated by rFu24 formulated in FΔF and LΔF, can the authors test higher serum dilutions in their GIA assay  ?

Author Response

We are grateful to the reviewer for the insightful comments on our paper. We have been able to incorporate changes to reflect most of the suggestions provided by the reviewer. We have highlighted the changes within the manuscript. A point-by-point response to the reviewer’s comments and concerns is attached.

Please see the attached document

Reviewer 2 Report

The manuscript by Anand et al. entitled "Production and immunogenicity of a tag-free recombinant chimera based on PfMSP-1 and PfMSP-3 using Alhydrogel and dipeptide based hydrogels" presents the recombinant production of a potentially immunogenic rFu24 protein derived from the fusion of two major vaccine target antigens, PfMSP-1 and PfMSP3, and its inclusion within a hydrogel matrix to enhance its immunogenic potential. Thus, the work has two main objectives, which concern the production of the recombinant, target-free protein and its entrapment into two hydrogels derived from two dipeptides, namely phenylalanine-α, and β-dehydrophenylalanine (FΔF) and Leucine-α, β-dehydrophenylalanine (LΔF). rFu24 protein enclosed into FΔF and LΔF hydrogels was then evaluated for its immunogenicity on BALB/c mice.

Some experimental sections are described in an insufficiently clear manner, preventing in my opinion their potential reproducibility. The lack of clarity of some sections does not allow me to define whether the conclusions are fully supported by the experimental data. This is true, for example, in the case of the ability of the two hydrogels to entrap the antigen.

I have listed the critical issues below, in the order in which I noted them in the text, and not in hierarchical order with respect to their impact on the quality of the manuscript.

Page 2, Based on similar strategy of using anti-malaria combinatorial drugs to avoid drug resistance [16. the approach of using subunit vaccines containing more than one malaria antigen or fusion protein chimeras-based blood-stage vaccine that tar-get multiple antigens may be more effective in limiting the parasite’s ability to escape host immunity and an enhanced chance of success [17].

The sentence is rather complex and not clear

Page 4, 2.2. Expression and purification of the full-length rFu24 .The colonies were grown in semi-defined medium containing kanamycin (30μg/ml) at 37°C with shaking till the culture reached an OD600 of 0.6 to 0.7.

It is unclear what the authors mean by "semi-defined medium”. The composition or a standard designation should be provided.

Page 4 The Q-Sepharose purified elutes.... Please substitute with "The Q-Sepharose eluates". In general, "elutes" should be replaced with "eluates"

Page 4 The antigenicity of rFu24 was confirmed by Western blotting using polyclonal antibodies against purified rFu24 raised in mice to confirm the identity of rFu24. Unclear and convolutedly constructed sentence.

Pag 5: The conformation integrity of the PfMSP-119 fragment in recombinant rFu24 was analyzed by enzyme-linked immunosorbent assay (ELISA) using conformation specific monoclonal antibodies 1H4 under nonreducing and reducing conditions as described earlier [30,31,67]. I wonder if, alongside the use of antibodies capable of recognizing conformational epitopes, it is not appropriate to apply a technique more generally suitable for the conformational study of proteins (i.e. CD or FTIR spectroscopy).

Pag 5: 2.5. Preparation of hydrogel: FΔF was dissolved in methanol at 50 mg/mL and LΔF in HFIP at 50 mg/mL using 10 min sonication. Instantaneous dipeptide hydrogels 1 wt %....

HFIP (I guess hexafluoroisopropanol) should appear at the first occurrence in the extended form.

Within the same paragraph, the three following expressions were used to designate hydrogel properties or amount: dipeptide hydrogels 1 wt %; 1% FΔF and LΔF gels; 1% (w/v) FΔF and LΔF hydrogels.

I am not an expert in this field, and I wonder whether “1 wt %” refers to the percentage of swelling or a strange way to refer to concentration (1 % w/v). In the former case, I wonder whether such a percentage of swelling that “instantaneously” characterizes the hydrogel is to be considered systematically obtainable by the procedure described here, or whether it was experimentally verified in the context of the present work.

Pag 5: 2.5 Entrapment of antigen. 25 μg of rFu24 was added to 1% (w/v) FΔF and LΔF hydro-gels at a ratio of 1:50 (Ag: FΔF/LΔF) by vortexing for 2hr at room temperature.

Due to the lack of clarity of the terms, it is also unclear to me how to interpret the procedure described for rFu24 entrapment.

Pag 5: Dipeptide gels of FΔF and LΔF at 1 wt% and rFu24 loaded FΔF and LΔF hydrogels as describe above.. Please substitute with "as described above". In general, the entire text would deserve a linguistic overhaul.

Pag 6: Parasite culture was maintained and synchronized as previously described by Trager and Jenson. A reference for this procedure should be given.

Figure 2 b, Were equal amounts of R and NR proteins loaded onto the western blot gel? Assuming equal amounts of protein were blotted on the filter, how can the Authors explain the more intense signal for R protein compared to NR?

Pag. 9 To study antigen entrapment in the hydrogels, 25 ug of rFu24 was added to 1% (w/v) FΔF and LΔF hydrogels at a ratio of 1:50 (rFu24: FΔF/LΔF) and incubated for one hour at room temperature.

As mentioned above, the procedure of antigen entrapment in the hydrogel matrix is not clear to me. If the protein is added to the pre-formed gel and preferentially adsorbed, I would observe that its complete entrapment supports the hypothesis of a strong chemical affinity with the matrix. What type of interaction would motivate this high affinity? How much general the obtained results?

Alternatively, the fact that no signal is observed in the supernatant (Fig. 4, a) could result from a strong dilution effect of the protein in the bulk where the entrapment is carried out. In this regard, another question arises: what is the dilution factor of rFu24 in the hydrogel bulk?

The hydrogels FΔF and LΔF were formed at 1 wt% peptide concentration and were self-supporting, colourless and translucent (Figure 3a, Supplementary Figure 2). I find self-supporting peptides is an unusual definition. Do the authors mean self-assembling peptides?

Fig.4: The caption should clearly state the nature and origin of the samples analyzed by SDS-PAGE. Please specify lane by lane.

The discussion is repetitive, with recurring conceptual modules. Merging and synthesizing these modules would help make it more concise and effective.

Author Response

(The authors gave the same response as above.)
